# Development of a Person-Centred Integrated Care Approach for Chronic Disease Management in Dutch Primary Care: A Mixed-Method Study

**DOI:** 10.3390/ijerph20053824

**Published:** 2023-02-21

**Authors:** Lena H. A. Raaijmakers, Tjard R. Schermer, Mandy Wijnen, Hester E. van Bommel, Leslie Michielsen, Floris Boone, Jan H. Vercoulen, Erik W. M. A. Bischoff

**Affiliations:** 1Department of Primary and Community Care, Radboud Institute for Health Sciences, Radboud University Medical Center, P.O. Box 9101, 6500 HB Nijmegen, The Netherlands; 2Science Support Office, Gelre Hospitals, P.O. Box 9014, 7300 DS Apeldoorn, The Netherlands; 3Pharos, Dutch Centre of Expertise on Health Disparities, P.O. Box 13318, 3507 LH Utrecht, The Netherlands; 4Research Group Innovation of Care and Services, HAN University of Applied Sciences, Kapittelweg 33, 6525 EN Nijmegen, The Netherlands; 5Department of Medical Psychology, Radboud Institute for Health Sciences, Radboud University Medical Center, P.O. Box 9101, 6500 HB Nijmegen, The Netherlands

**Keywords:** person-centred integrated care, person-centred care, chronic conditions, primary care, general practice, chronic care management, multimorbidity

## Abstract

To reduce the burden of chronic diseases on society and individuals, European countries implemented chronic Disease Management Programs (DMPs) that focus on the management of a single chronic disease. However, due to the fact that the scientific evidence that DMPs reduce the burden of chronic diseases is not convincing, patients with multimorbidity may receive overlapping or conflicting treatment advice, and a single disease approach may be conflicting with the core competencies of primary care. In addition, in the Netherlands, care is shifting from DMPs to person-centred integrated care (PC-IC) approaches. This paper describes a mixed-method development of a PC-IC approach for the management of patients with one or more chronic diseases in Dutch primary care, executed from March 2019 to July 2020. In Phase 1, we conducted a scoping review and document analysis to identify key elements to construct a conceptual model for delivering PC-IC care. In Phase 2, national experts on Diabetes Mellitus type 2, cardiovascular diseases, and chronic obstructive pulmonary disease and local healthcare providers (HCP) commented on the conceptual model using online qualitative surveys. In Phase 3, patients with chronic conditions commented on the conceptual model in individual interviews, and in Phase 4 the conceptual model was presented to the local primary care cooperatives and finalized after processing their comments. Based on the scientific literature, current practice guidelines, and input from a variety of stakeholders, we developed a holistic, person-centred, integrated approach for the management of patients with (multiple) chronic diseases in primary care. Future evaluation of the PC-IC approach will show if this approach leads to more favourable outcomes and should replace the current single-disease approach in the management of chronic conditions and multimorbidity in Dutch primary care.

## 1. Introduction

Over the last decades, the increasing prevalence of chronic diseases has cast a huge burden on healthcare systems worldwide [1]. Currently, chronic diseases are the leading cause of death globally, with cardiovascular diseases, diabetes, and chronic lung diseases causing the highest mortality [1]. In the Netherlands, 59% of the population had one or more chronic diseases in 2020 [2]. In addition, between 2004 and 2017, the prevalence of patients with two or more chronic diseases (multimorbidity) [3] in central Europe has increased in adults aged 50 and over [4]. Most importantly, chronic diseases have a major impact on patients’ health-related quality of life, especially when they have multiple chronic conditions [5,6,7,8].

To reduce the burden of chronic diseases on patients and healthcare providers, single disease management programs (DMPs) have been developed [9,10,11]. Based on Dutch primary care [12,13,14], we define DMPs as long-term chronic care programs in primary care that are predominantly run by general practice nurses (PNs) under the responsibility of a general practitioner (GP) and focus on assessing, monitoring, and treating a single chronic disease. DMPs for chronic obstructive pulmonary disease (COPD), cardiovascular diseases (CVD), and diabetes mellitus type 2 (DM2) are currently the most widely implemented. Although DMPs have shown some minor improvements in process indicators, such as coordination of care and communication between caregivers [15,16], they have failed to show improvement in patients’ health-related quality of life (HRQoL) [17,18]. A possible explanation could be that DMPs mainly focus on the medical aspects of a specific condition, with less attention being paid to other chronic diseases or social problems that may also impact HRQoL. In addition, an organisation in which patients with multiple chronic diseases attend multiple DMPs provided by multiple healthcare professionals (HCP) is not desirable, both from an economical and patient perspective [19]. Patients may receive overlapping or conflicting treatment advice [20]. Furthermore, the DMP approach seems to conflict with the core competencies of primary care professionals, i.e., medical generalism, community orientation, focusing on social determinants of health and societal factors, and working from a personal–professional relationship with patients [21,22]. 

An alternative approach for DMPs might be found in Person-Centred and Integrated Care (PC-IC), as increasingly advised by international guidelines on multimorbidity and chronic conditions [23,24,25]. Instead of focusing on a standard set of disease management processes determined by health professionals, PC-IC aims to ensure that patients’ values and concerns shape the way long-term conditions are managed [26]. This approach encourages patients to select treatment goals and to work with clinicians to determine their specific needs for treatment and support of their chronic diseases [27]. A PC-IC approach is believed to improve the quadruple aims [28,29] of better patient and HCP experience, population health, and cost-effectiveness [26,30]. Currently, several studies on such PC-IC approaches to managing chronic conditions in primary care are emerging, but descriptions of their scientific foundation are lacking [31]. 

In addition, in the Netherlands, a shift is taking place from DMPs to PC-IC approaches initiated by primary care HCP organizations. To scientifically support this movement, this paper describes a mixed-method multiphase development of a PC-IC approach for the management of patients with one or more chronic diseases in Dutch primary care. We co-designed the approach with all stakeholders involved, i.e., academics, HCPs, patients, and healthcare insurers. 

## 2. Materials and Methods

### 2.1. Design

A multiphase process to develop a PC-IC approach for patients with one or more chronic conditions, but at least DM, COPD, or CVD was started in March 2019 and finished in July 2020. We conducted the process together with three large primary care cooperatives in the eastern part of the Netherlands, i.e., the Nijmegen region (168 GPs, approximately 290,000 inhabitants), the Arnhem region (193 GPs, ~440,000 inhabitants), and the Doetinchem region (116 GPs, ~150,000 inhabitants). We followed a four-phase process in which the information collected in each phase was commented on by stakeholders and used in the next phase (see Table 1). The four subsequent phases were all a priori defined by the project team based on criteria for reporting the development of complex interventions in healthcare and including all relevant stakeholders [32,33]. In short, in Phase 1 we conducted a scoping review and a document analysis to identify key elements to construct a conceptual model for delivering PC-IC care. In Phase 2, national experts on DM2, CVD, and COPD and local HCPs commented on the conceptual model using online qualitative surveys. In Phase 3, patients with one or more chronic conditions commented on the conceptual model in individual interviews. To conclude the development process, in Phase 4 the conceptual model was presented to the local primary care cooperatives and finalized after processing their comments. We used the Standards for Reporting Qualitative Research (SRQR) guidelines to design and report the methods and results of the respective sub-studies [34]. 

The medical ethics review board of the Radboud University Medical Center declared that ethics approval for the study was not required under Dutch National Law (registration number: 2019-5756). All participants received written information about the study and their written informed consent was obtained prior to their participation.

### 2.2. Scoping Review and Document Analysis (Phase 1)

In this phase, we aimed to identify which process elements and which interventions a PC-IC approach should contain. We identified the key process elements (e.g., history taking or discussing patients’ goals) for successful (multiple) chronic disease management by conducting a scoping review. We identified the key interventions by conducting a document analysis.

For the scoping review, we searched PubMed, EMBASE, Cochrane, Turning Research into Practice (TRIP) Medical Database, and the Guidelines International Network (GIN) to identify key elements for the successful management of (multiple) chronic diseases in primary care (see Appendix A for the search strategies). All eligible publications up until 27 August 2019 were included, and no lower limit with regard to publication date was applied. Forward citation tracking was used and the reference lists of relevant publications were hand searched for additional relevant publications. Two of the authors (LR and MW) independently screened the titles and abstracts of the publications and reviewed the full text of those that seemed eligible for the scoping review. Publications were included if the language was English or Dutch, if the target population consisted of patients with multiple chronic conditions, and if the target setting was primary care. Primary care was defined as a non-hospital community setting with medical care continuity by (the equivalent of) a GP. Publications were excluded if they were study protocols, commentaries, or cost-effectiveness analyses. Next, one author (LR) extracted data on publication details, methods used, and recommendations on important elements of clinical care from the included publications. The extracted details were cross-checked by a second author (MW). The results of the scoping review were used to create a conceptual model including key process elements for PC-IC.

For the document analysis, we analysed all Dutch chronic disease care standards and GP guidelines relevant to the DMPs for COPD, CVD, and DM2 [12,13,14,35,36,37] to identify all unique interventions that were used in the management of these conditions. The documents were analysed by two authors (LR and MW) using inductive thematical coding (Table 2). Using an affinity diagram, a schematic overview of unique key interventions to be included in the PC-IC approach was developed. The resulting intervention model was combined with the process model from the scoping review to form our conceptual PC-IC approach, which was further adjusted in the subsequent phases.

### 2.3. Online Surveys with Healthcare Professionals (Phase 2)

We conducted online surveys among healthcare professionals using open-ended responses, with a thematic analysis of wordings in order to further adjust the conceptual model of our PC-IC approach. This method was chosen because it enabled HCPs from different disciplines to give their individual opinions and flexibility to contribute to the study at a time that suited participants. Each regional primary care cooperative purposively selected a heterogenous group of 10 to 15 HCPs in the following professions or disciplines: GPs with a special interest in CVD, DM, or COPD, regular GPs, PNs, allied HCPs (e.g., physiotherapists, dieticians), social workers and other HCPs involved in the care for patients with chronic diseases. In addition, six GPs with a special interest in CVD, DM, or COPD who were involved in the national guidelines or health policy committees were asked to participate. All participants were monetarily compensated for their time and received written information on the conceptual model of the PC-IC approach before the online survey started.

The online survey was performed in five subsequent parts in which open-ended questions were sent to participants through an adapted secured version of LimeSurvey (LimeSurvey GmbH, Hamburg, Germany). Each survey focused on a predetermined part of the conceptual model of the PC-IC approach. Questions concerned the strength and limitations of different parts of the PC-IC approach. If there were doubts about the responses to the questionnaire items, we asked follow-up questions via e-mail or phone until the answers could be sufficiently interpreted. Analysis of the questionnaire data was performed by three researchers (LR, MW, and AO) using thematical coding, as described in Table 2. To conclude this phase, we organized a virtual meeting with all participants in which we presented the results of the surveys and checked for agreement. This resulted in an adapted version of the conceptual model of the PC-IC approach.

### 2.4. Individual Interviews with Patients (Phase 3)

We then organized individual semi-structured telephone interviews with chronic disease patients to explore their opinions on the conceptual model of the PC-IC approach. Each primary care cooperative recruited patients with DM2 and/or COPD and/or CVD who received chronic disease management from their general practitioner. Participating patients received written information on the study and the conceptual model of the PC-IC approach by e-mail or postal mail before being interviewed. Patients were recruited until data saturation was reached. Patients did not receive financial compensation for their participation. 

The interviews were conducted by two researchers (LR and FB). The interviewer first explained the goal of the interview and presented the conceptual model before asking questions regarding expected strengths, weaknesses, and points for improvement of the different elements and interventions (see Appendix B). The interviews were audio recorded, transcribed verbatim, coded, and analysed according to the thematic analysis approach, see Table 2. A summary of the results was offered for member checking. This resulted in an adapted version of the conceptual model of the PC-IC approach. 

### 2.5. Finalization of PC-IC Approach (Phase 4)

In this last phase of the development process, we aimed to collect final feedback from the remaining stakeholders (see Table 1) on the adapted version of the conceptual model of the PC-IC approach. Because of their vital role in the organisation and reimbursement of primary healthcare for chronic patients, representatives of the three primary care cooperatives involved and three healthcare insurance companies were invited to and participated in a joint meeting to give oral feedback on the adapted version of the PC-IC approach from their perspectives. Neither patients nor HCPs were invited to this meeting. After the presentation of the PC-IC approach by one of the authors (LR) an open discussion with the ten participants was moderated by another author (EB). Notes were taken by one of the authors (LR) during the discussion.

Finally, to improve the comprehensibility of the approach for people with limited health literacy, two experts from the Dutch Centre of Expertise on Health Disparities (Pharos) were asked to provide written feedback on the comprehensibility of the conceptual model. Their feedback was collected and summarized by one of the authors (LR).

All input from phases one through four was processed by the research team in a report of the feedback on the PC-IC approach. This report was shared with the participants and a meeting was held with stakeholders of the primary care cooperatives for the finalization of the PC-IC approach. 

## 3. Results

### 3.1. Scoping Review and Document Analysis (Phase 1) 

#### 3.1.1. Scoping Review

We identified 203 unique publications, of which 18 were included in the review (Table 3). Included publications were published between 2007 and 2019, of which 67% were in the last five years (2015–2019). All publications were in English and most were from the United States or the United Kingdom. 

Most publications stated there is still a lack of research and thus insufficient evidence for optimal clinical management of people with multiple chronic diseases [5,23,38]. Only a few of the included studies focused on person-centred outcomes [38,39]. Nonetheless, authors generally agreed that interventions that are generic in nature (i.e., not specific for the underlying condition(s)) and with a person-centred approach are most likely to result in health benefits for patients with chronic diseases and multimorbidity, in comparison to a single disease approach [5,39,40,41].

##### Assessment of Multiple Domains—Integral Health Status

Besides the medical domain, authors recommended paying attention to other domains of life as well, i.e., to functional limitations, mental health, and social functioning [5,24,39,40,41,43,47,48,50,51]. Patients with limited physical, emotional, and financial capacities are most disrupted by their chronic illness, but interventions to support these particular patient capacities have been scarcely studied [39]. With regard to mental health, it is recommended to discuss this domain with patients and to actively monitor signs of anxiety, distress, and depression [24,47]. For the social domain, social circumstances, including social support, living conditions, and financial constraints should be considered [47]. Health professionals are encouraged to involve relatives or other informal caregivers in key decisions about the management of the patient’s health, if the patient so desires [24,40,48]. In addition, the needs of these relatives should be considered as well [41]. By including all of these domains, interventions have the potential to better address health inequalities in the population [50]. We summarized the multiple domains in the concept of integral health status (Figure 1).

**Table 3 ijerph-20-03824-t003:** Details of the publications included in the scoping review; publications are listed in alphabetical order of the authors.

Authors	Country	Type of Publication	Title
AHRQ—American Geriatrics Society Expert Panel on the Care of Older Adults with Multimorbidity 2012 [23]	United States	Consensus document	Guiding Principles for the Care of Older Adults with Multimorbidity: An Approach for Clinicians
Boehmer & Abu Dabrh et al., 2018 [39]	United States	Systematic review	Does the chronic care model meet the emerging needs of people living with multimorbidity? A systematic review and thematic synthesis
Boehmer & Guerton et al., 2019 [42]	United States	Descriptive article	Capacity Coaching: A New Strategy for Coaching Patients Living with Multimorbidity and Organizing Their Care
Boyd & Fortin 2010 [40]	United States	Review	Future of Multimorbidity Research: How Should Understanding of Multimorbidity Inform Health System Design?
Culpepper 2012 [43]	United States	Review	Does Screening for Depression in Primary Care Improve Outcome?
Engamba & Steel et al., 2019 [44]	United Kingdom	Analysis	Tackling multimorbidity in primary care: is relational continuity the missing ingredient?
Fortin & Hudon et al., 2007 [45]	Canada	Review	Caring for Body and Soul: the Importance of Recognizing and Managing Psychological Distress in Persons with Multimorbidity
Hopman & de Bruin et al., 2016 [38]	Netherlands	Systematic literature review	Effectiveness of comprehensive care programs for patients with multiple chronic conditions or frailty: A systematic literature review
Lenzen & Daniëls et al., 2015 [46]	Netherlands	Background paper	Setting goals in chronic care: Shared decision-making as self-management support by the family physician
Marengoni & Angleman et al., 2011 [5]	Sweden	Systematic literature review	Aging with multimorbidity: A systematic review of the literature
Muth & van den Akker et al., 2014 [47]	Germany	Original study	The Ariadne principles: how to handle multimorbidity in primary care consultations
Muth & Blom et al., 2019 [48]	Germany/United Kingdom	Systematic guideline review & expert consensus	Evidence supporting the best clinical management of patients with multimorbidity and polypharmacy: a systematic guideline review and expert consensus
NICE—National Institute for Health + Care Excellence 2016 [24]	United Kingdom	Guideline	Multimorbidity: clinical assessment and management
Poitras & Maltais et al., 2018 [41]	Canada	Scoping review	What are the effective elements in patient-centred and multimorbidity care? A scoping review
Ricci-Cabello & Violan et al., 2015 [49]	United Kingdom	Scoping review	Impact of multi-morbidity on quality of healthcare and its implications for health policy, research, and clinical practice. A scoping review
Smith & Wallace et al., 2016 [50]	Ireland	Systematic review	Interventions for improving outcomes in patients with multimorbidity in primary care and community settings
Stokes & Man et al., 2017 [51]	United Kingdom	Review	The Foundations Framework for Developing and Reporting New Models of Care for Multimorbidity
Wallace & Salisbury et al., 2015 [19]	United Kingdom	Review	Managing patients with multimorbidity in primary care

##### Case Management

Case management is considered to be an effective way to support patients in achieving their goals and communicating with other HCPs [41]. Case managers are advised to perform regular face-to-face assessments with the patient [41]. Establishing a partnership between different disciplines (i.e., primary care physicians, medical specialists, nurses, mental health professionals, and social care workers) may provide the key to improving care for patients with multimorbidity and psychological distress [45,48]. The patient should also be part of this team [41]. Communication and coordination across health professionals are considered essential in providing multimorbidity care [24,39,40,47,48,49]. To improve partnership and communication between health professionals and the patient and family, it is recommended to work in small teams with dedicated contact persons on both sides [44].

##### Clinical Assessment

Multiple publications recommend assessing disease burden by determining how day-to-day life is affected by the patient’s health problems and establishing how health problems and treatments interact [24,48]. Examples of health problems influencing disease burden are chronic pain, depression and anxiety, and incontinence [48]. Another recommendation is to assess the burden of treatment because this can greatly influence patients’ quality of life [23,24,39,42,47,48]. For example, NICE recommends discussing the number of healthcare appointments a patient has and the format in which they take place, the number of non-pharmacological treatments, the assessment of polypharmacy, and the effects of all treatments on mental health or well-being [24]. An annual medication review is recommended to evaluate the risks, benefits, possible interactions, and treatment adherence for each drug the patient uses [24,48]. Finally, Muth et al. noticed that the management of risk factors for future disease can be a major treatment burden for patients with multimorbidity and should be carefully considered when optimizing care [48]. 

##### Patient Preferences and Priorities

Many studies described the importance to elicit patients’ preferences and priorities for care [23,24,40,42,45,47,48,50]. Addressing a patient’s priorities helps to minimize adverse effects of psychological distress [45]. Using these preferences and priorities, together with the health professional’s clinical expertise and based on the best available evidence, individual goals for care should be determined [46,47,48]. In this conversation, health professionals should also explore, without any assumptions, to what extent a patient wants to be involved in decision-making [48]. Another important factor to consider when discussing goals with patients with multimorbidity is life expectancy and prognosis of the conditions [23].

##### Care Plan

After prioritizing the patient’s problems, a care plan should be drafted, which sets out realistic treatment goals, monitoring, treatment, prevention, (self-)management advice, responsibility for coordination of care, and timing of follow-up through shared decision-making [24,47,48]. The plan should be shared with other involved professionals, the patient, and the family [41,44]. When choosing interventions, it is advised to use the best available evidence, but to also recognize the limitations of the evidence base for patients with multimorbidity [23,48] and to check if an intervention is effective in terms of patient-related outcomes [24]. Possible interventions should be tailored and adapted to a patient’s individual needs [41,50] and shared decision-making should be used to maximize the impact of interventions [19,40,41]. The key process elements of the PC-IC approach that we could retrieve from the included publications are summarized in Figure 2.

#### 3.1.2. Document Analysis 

For this phase, we analysed three clinical guidelines of the Dutch College of General Practitioners and three national care standards for DM2, COPD, and CVD [12,13,14,35,36,37]. The document analysis resulted in a list of categories with unique key interventions for disease-specific and holistic care (Appendix C) which was converted into a draft conceptual intervention model for the PC-IC approach. After processing feedback from stakeholders as described in Section 3.2, Section 3.3 and Section 3.4 below, this resulted in a graphical representation of the final intervention model for use in daily practice.

### 3.2. Online Qualitative Surveys with Healthcare Professionals (Phase 2)

A total of 56 HCPs were invited to participate in the online qualitative survey study. Fifty-two (93%) responded and 10 were asked follow-up questions to clarify the responses of their initial input. The majority of the participants consisted of GPs (n = 16) and PNs (n = 15), but several other disciplines were also involved (Table 4). The results of the survey were categorized as: general comments on the PC-IC approach and comments on the individual phases of the care process (i.e., assessment; setting personal health goals; choosing interventions; individual care plan; evaluation).

#### 3.2.1. General Comments

In general, most participants agreed with the underlying vision of the PC-IC approach, namely that person-centred and holistic care would improve the quality of care for patients with one or more chronic diseases (Q1, see Table 5). It is likely to lead to more insight into the patient’s health status and any underlying problems. Using the PC-IC approach could increase the motivation of the patient for behavioural change and therefore may improve therapy compliance and health status. Many participants expect that this approach will initially take up more time, but that this time will be restored in the future. In the long term, therefore, the approach could save time and lead to more efficient provision of care (Q2).

Another anticipated advantage of the PC-IC approach is the cyclical aspect, which ensures that the process continues and the patient’s health status is checked repeatedly. Some participants liked the fact that the PC-IC approach has a strong theoretical basis and would give patients more control and responsibility (Q3). 

According to some participants, a potential disadvantage of the approach could be that it may be too time-consuming, both for the HCP and for the patient (Q4). Therefore, some participants considered it not feasible to implement the approach in daily practice in its current form. In addition, some participants doubted the magnitude of the positive effects on the quality of care and patients’ health of the PC-IC approach. 

In addition, participants questioned which patients the care program would be suitable. Some thought it would be useful for all patients, whereas others suggested using it only for the more complex patients. Others indicated that the program may be too complicated for people with limited health skills (Q5). 

#### 3.2.2. Assessment of Integral Health Status

Assessing patients’ integral health status was considered a positive development by almost all participants, who indicated that a broader assessment of health status may have positive effects for both the patient and the HCP. It provides insight into the connection between health problems and their underlying causes for both parties. This creates more awareness and motivation for change in patients, especially if the underlying cause of these health problems concerns a domain other than the medical domain (Q6). Involving family members or informal caregivers in discussing the overall health situation was also mentioned as a strong point. They can often provide useful additional information and may be supportive during treatment. Assessing and discussing integral health status also provides clear goals and priorities for the patient. Therefore, participants considered the integral health status a suitable way to map out complex patients. Filling out a questionnaire online (at home) helps the patient better prepare and saves time during the consultation.

A disadvantage of focusing on integral health status instead of the disease-oriented approach might be that the medical aspects may not be sufficiently addressed and the severity of individual chronic diseases becomes less clear to patients. In addition, HCPs feared a patient may not want to talk about other areas of life as he/she may consider them irrelevant to the condition. It was also mentioned that making a more elaborate assessment of the patient’s health status could be confrontational for some, especially for those with many problems (Q7).

#### 3.2.3. Setting Personal Health Goals

Most participants were enthusiastic about setting personal health goals through shared decision-making. The most important advantage mentioned was that it may motivate the patient toward behavioural change. Contributing factors to motivation were awareness, commitment, and responsibility on the side of the patient. Setting personal goals also benefits the HCP, who gains more insight into the patient’s priorities, and is more in tune with the patient, which could make interventions more effective (Q8). Participants also mentioned the disadvantages and pitfalls of setting personal goals, such as that the importance of disease control might be overlooked (Q9). In addition, HCPs mentioned the risk that the patient sets unattainable goals, which can demotivate both the patient and the HCP.

#### 3.2.4. Choosing Interventions

Although the graphical representation of the PC-IC conceptual model for use in daily practice and the accompanying schematic overview of existing key interventions to support the management of patients with chronic conditions (see Section 2.2) was appreciated by many participants, the graphical representation in its initial form as presented to the participants was deemed confusing by some of them due to the inclusion of too much information in one visualisation. Without further explanation, this makes the model difficult to understand (Q10). Participants also mentioned that it is difficult to create a static model for the supply of interventions, which will usually vary between regions and possibly change over time.

#### 3.2.5. Individual Care Plan

Participants saw many advantages of a care plan, both for the patient and for the HCP. The most important advantage is that the patient and the various HCPs involved may share the same specific personal goals, which makes communication between patient and caregiver and between different caregivers easier. The care plan provides a clear structure and benefits evaluation of personal goals. Participants also indicated that it fits well within a holistic approach (Q11). Disadvantages could be that it is time-consuming to draw up the plan, that the conversation with the patient can become subordinate to the plan, that making a plan is not yet sufficiently integrated into the ICT systems, and that a care plan can lead to the medicalisation of non-somatic problems. Disadvantages for patients could be that it can be invasive, that it can evoke resistance, and that it can create ambiguity if not all HCPs are on the same page (Q12).

Participants wanted to include the following information in the proposed individual care plan format: the patient’s specific goals; the selected interventions; an overview of the HCPs involved and their responsibilities; and time of evaluation.

#### 3.2.6. Evaluation

Many participants found it unclear whether a patient-level evaluation had been included in the process of the PC-IC approach. They indicated that they missed this essential step and would prefer to add it (Q13). An advantage of an evaluation is that it provides new information which can be used in the next cycle. No disadvantages of an evaluation were mentioned.

### 3.3. Individual Interviews with Patients (Phase 3)

Twelve patients were invited for the interview study. One patient did not want to participate, two were not eligible as they did not receive care in a DMP, and nine consented to be interviewed. Data saturation was reached after the first eight interviews. Eight patients (88.8%) were male, their mean age was 65 years (range 58–79 years). One patient had COPD, three had CVD, two had DM2, and three had any combination of these chronic diseases. The median duration of being in the DMP was ten years (range 2–10 years).

The following main categories were”Ide’Iified during the analysis of the interview transcripts: personalized care, cooperation, patient role, and PN role. 

#### 3.3.1. Personalized Care

Patients were generally positive towards the presented manner of personalized care, especially regarding the integral health status assessment and use of individual care plans. The integral health assessment may give patients and HCPs better insight and focus on holistic well-being, and may account better for comorbidity, disease interaction, and psychological factors. It may detect issues affecting patients’ well-being and identify those who require more support. It may also improve working relationships by shifting towards a more personal approach rather than a disease focus. Two participants were content with their current care and expected no benefits from the new approach. 

The PC-IC approach would be inviting patients to be more involved in their healthcare. Having the care plan at home could help remind and motivate them and allow easier involvement of informal caregivers/social support systems. The wording of the care plan should be easy to understand. Some participants called for more flexibility regarding individual care plans to be adaptive to patients’ needs and unexpected circumstances and requested options to communicate their questions and concerns to their PN after the plan is formulated.

Longer consultation time would allow for more personal attention, and opportunities for better patient education, and is expected to benefit health outcomes. One participant believed that too much consultation time was reserved for patients. Some participants suggested adjusting the consultation frequency and the duration according to each patients’ individual needs. This may allow PNs to direct their efforts more efficiently (Q14).

#### 3.3.2. Co-Operation

Participants saw the benefits of being equal stakeholders in their own care. This may improve care participation and help them carry responsibility for their own health. Greater equality may also improve the working relationship with HCPs. Giving patients the opportunity to prepare for care consultations was seen as a way to improve participation and equality. Using digital questionnaires for assessing integral health status was appreciated by the participants. Avoiding time constraints when answering health-related questions may also cause more reflection on health, better quality answers, and time during consultations to explore the answers. One participant noted that completing the questionnaire allows patients to share thoughts about their health with their informal caregivers/social support system more easily (Q15).

Some participants worried that patients with low literacy, facing language barriers, insufficient health skills, or insufficient computer skills may have difficulties using the questionnaire. One participant affirmed this, saying his low literacy made him feel insecure and uncertain when filling out questionnaires. One participant thought that thirty minutes was too long to fill out a questionnaire. Participants gave several suggestions regarding accessibility. Intelligible and straightforward questions were seen as important. Further suggestions were: visual instead of numeric response scales, a paper-version alternative, and a narrator function. One participant suggested a shorter alternative to the questionnaire.

Using the questionnaire results would support the patient and the PN, as both may get better insight into the patient’s current health status and its long-term course. This may provide a sense of control and assurance for the patient and may help both sides to prepare for consultations and help discover previously unacknowledged problems that affect the patient’s health when the responses yield unexpected results. Several participants mentioned that the color-coding of the results made them more insightful, while one participant found this too confrontational and judgmental towards patients.

Several participants saw potential flaws in using the questionnaire results. Two participants warned that this could lead to a search for non-existent problems. One participant thought paying attention to psychological stressors was lacking, while these can cause or amplify illness. Participants mentioned that the results should also be kept simple, some suggesting that a summary would suffice. Some participants suggested additional questions, one suggesting a question about literacy, another suggesting to include socioeconomic background, and a third suggesting questions regarding what mattered most in the life of patients (Q16). 

Participants also provided advice on the quality of communication with their HCPs, from which five requirements for communication emerged: trust, authenticity, empathy, constructiveness, and specificity. Trust improves patient openness and working relationships and requires continuity of care. Authentic personal interest makes patients feel seen and heard, and is conducive to developing trust. Being empathetic may provide a sense of safety and comfort, and let patients know that they are supported. Being constructive may create a positive and motivational focus on the patient. Finally, adjusting one’s approach to specific patient abilities and needs may benefit mutual understanding and working relationships.

#### 3.3.3. Role of the Patient

Participants thought that gaining ownership and self-management was important and that patients are ultimately responsible for their actions regarding their well-being, but may often be unaware of their potential influence on it. Being aware of this may stimulate self-management. They noted the potential benefits of self-management but also expressed thoughts on factors limiting its attainability. Participants thought that experiencing ownership in care may motivate better adherence to treatment and healthcare advice, and may facilitate acceptance of advice. They noted that any level of self-management may be beneficial for this. Similar to the HCPs the participating patients also mentioned that formulating personal health goals may contribute to more personalized care. Patient-specific factors such as personality traits, acceptance, and knowledge were thought to have an important limiting influence on attaining self-management (Q17). Several participants noted that patients’ responsibility extended to communication with the PN, as patients may choose to withhold information on topics such as mental problems or illiteracy, but this may prevent them from receiving optimal care.

Three participants elaborated on involving informal caregivers/social support systems or primarily spouses, during consultations and at home. Patients bringing their spouses to consultations may be a source of information for the PN. The spouse may help retain information and provide support at home, as well as develop more understanding of the patient and their problems themselves. One participant noted that this involvement should be balanced with professional care, as a patient might value the opinion of their spouse more than that of the PN (Q18).

#### 3.3.4. Role of the Practice Nurse 

Participants saw benefits in the proposed role of the PN in providing a patient-centred model of care, but also mentioned several limiting factors and provided feedback on their perception of PNs’ responsibilities in the process. The PN taking on the case-manager role may provide a more central viewpoint of patient wellbeing, in line with the integral health assessment. Having a central point of responsibility may also benefit from continuity of care. However, some PNs may lack the knowledge and skills to deal with complex cases or the affinity to handle certain aspects of patient well-being. Guiding patients toward appropriate care when faced with these limitations was marked as a responsibility of the PN. One participant noted that patients may still prefer GP visits for certain problems regardless of the PN’s capability. Resource constraints, such as available time per patient, were seen as a potential limitation. 

Other PN responsibilities mentioned were on supporting self-management and communications within the healthcare team. Participants noted that improving self-management is dependent on the PN creating opportunities to do so, which might require them to develop flexibility in their approach according to patients’ needs. Two participants suggested that HCPs could provide summaries of consultations as additional support when formulating personal goals. Sharing of relevant information between HCPs was seen as important in keeping care teams informed on patient health and may prevent patients from having to repeat their story several times.

Participants had different opinions on GP involvement in their care. Most participants advised that their GP did not need to be ‘visibly’ involved in their care, one saying that he expected the PN to have more relevant expertise than the GP, and another saying that the GP should be involved when deemed required. One participant saw merit in some but limited GP visibility, even if only once a year (Q19).

### 3.4. Finalization of Recommended PC-IC Approach (Phase 4)

#### 3.4.1. Health Insurers

In general, health insurers found the PC-IC approach a good and positive development to move towards integral and holistic tailor-made care. Their suggestions for further improvement were: to describe the inclusion criteria for the PC-IC approach in practice more clearly, for example, every patient with two or more chronic diseases; pay more attention to the consequences of a shortage of HCPs in the future by having patients prepare their consultation at home and using more e-health applications; pay more attention to the required change in organisations and practices, because the implementation of the intervention determines if the intervention is successful.

#### 3.4.2. Dutch Centre of Expertise on Health Disparities (Pharos)

The experts from Pharos felt positive about the PC-IC approach to health and treatment because they found that, for a lot of people in vulnerable positions, not only disease, but also context, abilities, and possibilities influence health. A digital questionnaire for the assessment of health status that is already used in Dutch hospitals and general practices (the Nijmegen Clinical Screening Instrument, or NCSI) [52] was tested with people with limited health skills and led to suggestions for improvement of the language use and layout of this digital questionnaire. In addition, Pharos provided feedback on the conceptual intervention model, which was found to be an unsuitable way to visualise and discuss these treatments. The model was considered too complicated and interfered with the integral approach. 

#### 3.4.3. Finalization of the PC-IC Approach

Based on the scientific literature, current practice guidelines, and input of a variety of stakeholders, the holistic, PC-IC approach for the management of patients with (multiple) chronic diseases in primary care was finalized in a meeting with relevant stakeholders of each primary care cooperative (Figure 3 and Figure 4). 

## 4. Discussion

### 4.1. Summary of Results

To our knowledge, this paper is the first to describe in detail the subsequent steps in the development of a person-centred and integrated care approach for people with (multiple) chronic conditions in primary care. In the first phase, the scoping review identified that a PC-IC approach for multimorbidity should comprise multiple domains of health status, a case manager, and a thorough assessment of patient preferences and priorities. These essential elements were incorporated into a conceptual model for the PC-IC approach. The document analysis resulted in a list of unique interventions. In the second phase, HCPs commented on the (dis)advantages of the conceptual model, and provided suggestions for the improvement of the conceptual intervention model. The third phase consisted of a patient-level evaluation step to the PC-IC approach. Patients commented on the conceptual model and indicated that this approach could have many advantages, such as being more responsible for their own health and having a partnership with the HCP. In the final phase, health insurers and the Dutch Centre of Expertise on Health Disparities (Pharos) provided feedback on the model, after which the PC-IC approach was finalized in a meeting with relevant stakeholders of each of the three primary care cooperatives involved.

### 4.2. Comparison to Existing Literature & Interpretation 

Our findings are supported by other interventions to deliver personalized primary care for patients with chronic conditions that have been reported [31,53,54]. Similar to our approach, these interventions all include a PC-IC consultation, case management, personal goal setting, and network support. Differences between the respective approaches consist mainly of the targeted population and the way eligible patients are selected. The most recent interventions focus on targeting multimorbidity or ‘high-need’ patients. For example, Salisbury et al. developed and evaluated the 3D approach for people with multimorbidity in the UK, in which general practices offered greater continuity of care and biannual person-centred, comprehensive health reviews [31]. They selected patients with at least three types of chronic diseases and, although patients experienced the provided care as more person-centred, no favourable effects on HRQoL, general well-being, or patients’ treatment burden were observed [31]. Another intervention, which was also developed in the Netherlands, divides patients into low-, moderate-, and high-care-need subgroups, and only the high-care-need subgroup receives the intervention [55,56]. The effects of this intervention have not been reported yet, but a likely advantage of targeting all patients with chronic conditions, as we aim with our intervention, is that it may reduce overtreatment in patients who actually need less care than they currently receive according to the strict DMP protocols. This may create more time for patients who need more attention from their primary care HCPs. 

The results from our interviews with patients suggest that the developed PC-IC approach may solve several problems in current chronic care. For example, Rimmelzwaan et al. found that people with multimorbidity missed an approach that focuses on the patient “as a whole” [57]. In addition, these authors also observed that the participants in this study reported that HCPs should treat their patients as equals. Our study shows that patients believe that this new PC-IC approach could improve holistic care, time, and attention in consultations with the NP, as well as the partnership between patients and HCP. Furthermore, our findings are similar to research by Rijken et al. [58], who found that people with multimorbidity have the following priorities in their chronic care: having one health record shared by all HCPs involved in their care, regular comprehensive assessments, and receiving support from their HCPs to self-manage their chronic conditions.

In our study, we have predominantly focused on the micro-level service delivery aspects of PC-IC care. However, to support the PC-IC approach, other levels, and components of integrated care, i.e., the meso and macro levels of service delivery, leadership and governance, workforce, financing, technologies and medical products, and information and research, have to be considered and studied as well [10,59]. For financing, Bour et al. have studied a complementary payment model to this PC-IC approach, which is published elsewhere in this journal [60].

### 4.3. Strengths & Limitations

A particular strength of our study was the rigorous and extensive development process per region with relevant stakeholders. The development of the PC-IC approach based on the existing literature and the input from stakeholders makes the foundation of the conceptual model the best it can be before the scheduled feasibility study is executed, making the feasibility study even more effective. Because the development process was finalized per region, it could be tailored to fit the regional situation. We did, however, not further analyse regional differences, which limits the generalisability of the results to other regions in the Netherlands or other countries. Another advantage of our study was that HCPs and patients could comment on a tangible conceptual model, which made their feedback more specific and useful to modify the concept. A final strength of the study was the high participation rate of HCPs. This may be due to the method of online interviews, because of the advantages of online interviewing: significant savings in time for participants and the opportunity for participants to carefully formulate a response to a particular question [61]. Another explanation could be the compensation HCPs received from the regional primary care cooperatives to participate in the study. 

We also acknowledge some limitations. First, in the beginning of the project we performed a scoping review on multimorbidity, but the scope of the project later expanded to people with one or more chronic diseases, also because of the feedback from participating HCPs. Nonetheless, we think the findings are also relevant for patients with single chronic diseases, as problems may still arise in other areas of life, and PC-IC seems also effective in single disease cases [26]. In addition, the scoping review is currently somewhat outdated. However, we decided not to update the scoping review at this stage, as the intervention is based on the consecutive phases of the development process. 

Second, in the interview study (Phase 3), eight of the nine patients interviewed were males, which limited our ability to take the role of gender into account when adapting the draft conceptual PC-IC model from the patient perspective. This clearly reduced the diversity of the study sample and may also explain why data saturation was reached rather quickly.

Third, due to the influence of COVID-19 restriction measures, the method of interviewing patients had to be revised. To limit the potential exposure of patients with chronic diseases to the SARS-CoV-2 virus, we chose to conduct the interviews by phone. The pitfall of this method is that non-verbal signals cannot be seen, which might lead to different conversations and different observations from the interviews. An advantage might be that the patient feels more anonymous and is more likely to respond frankly, although the topic of our study was not particularly sensitive. 

Fourth, HCPs and patients commented on a theoretical model. After actually experiencing it in their practices, their views and opinions may be different. Therefore, the experiences of patients and HCPs should also be examined after having implemented the model in the upcoming feasibility study.

### 4.4. Implications

#### 4.4.1. Recommendations for Future Research

Our next studies will focus on the feasibility and the actual effects of the developed PC-IC approach in terms of the Quadruple Aim, in which we will focus on health-related quality of life, self-management behaviour, and patient experience, as outcome variables in research on the effects of PC-IC should be tailored to be person-centred [62]. As part of the cluster, in the randomised trial that is currently underway we assess barriers and facilitators of switching from the current to the new (PC-IC) approach in several domains (i.e., professionals, patients, organizational, and financial domains). The insights we gain from this will be part of the recommendations regarding the implementation of the PC-IC approach elsewhere. Furthermore, more research is needed on the acceptability of this approach in patients with limited health literacy. 

#### 4.4.2. Recommendations for Practice

Although this study offers some important insights for HCPs searching for a PC-IC approach to chronic care, the anticipated superiority of this approach relative to the current DMPs has yet to be studied.

## 5. Conclusions

Based on the scientific literature, current practice guidelines, and the input of a variety of stakeholders, we developed a holistic, person-centred and integrated approach for the management of patients with (multiple) chronic diseases in primary care. Future evaluation of the PC-IC approach will show if this approach leads to more favourable outcomes and should replace the current single-disease approach in the management of chronic conditions and multimorbidity in Dutch primary care.

## Figures and Tables

**Figure 1 ijerph-20-03824-f001:**
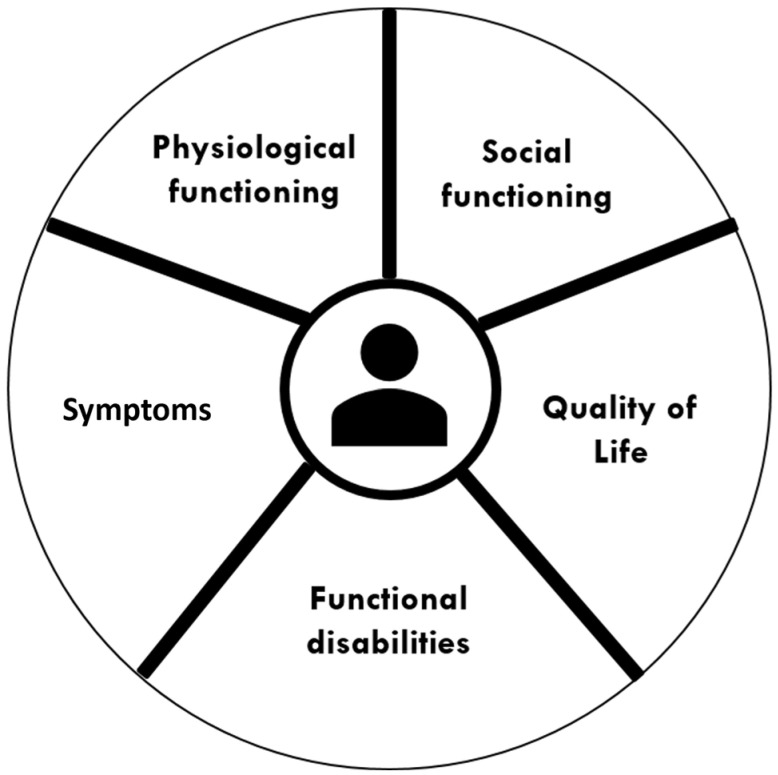
Domains in the concept of integral health status.

**Figure 2 ijerph-20-03824-f002:**
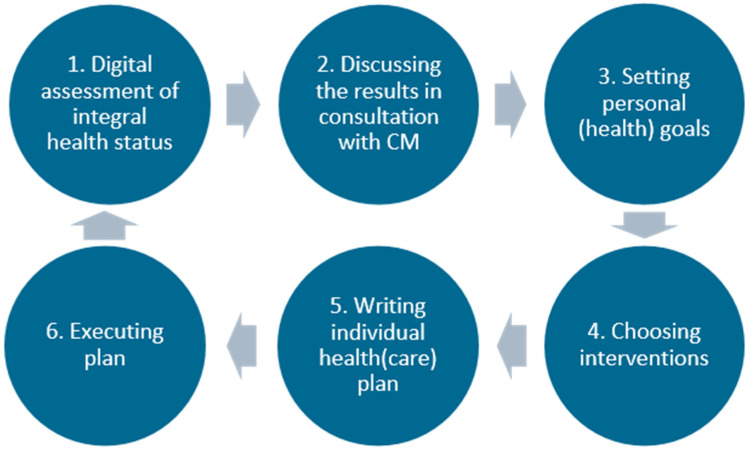
Conceptual model for PC-IC for patients with (multiple) chronic conditions in primary care. In the Dutch primary care setting, the PN in the general practice was recommended to serve as case manager. They can consult the general practitioner or other HCPs when necessary. The first step in this program is assessing the integral health status of the patient (health across multiple domains), using a (preferably digital) questionnaire at home and physical measurements. The second step is an appointment in which the results are discussed with the patient in a semi-structured way. The case manager discusses if the results are recognizable, if there are other issues that have not come up, and the priorities of the patient. Personal goals are formulated in the third step, which can range from purely medical goals to social goals. In the fourth step, the HCP and patient will together choose the right interventions to achieve these goals, from the experience of the HCP, the ideas of the patient, and a list of regional options. The goals and interventions are documented in a personal healthcare plan, which is preferably digitally available for all involved HCPs and the patient. Next, referrals are made if necessary and the treatment is started. An evaluation is planned and carried out, multiple times if necessary. If a treatment goal is reached or another treatment goal is more urgent, the cycle can be repeated. Abbreviations: CM: case manager; HCP: healthcare provider; PC-IC: Person-Centred and Integrated Care; PN: practice nurse.

**Figure 3 ijerph-20-03824-f003:**
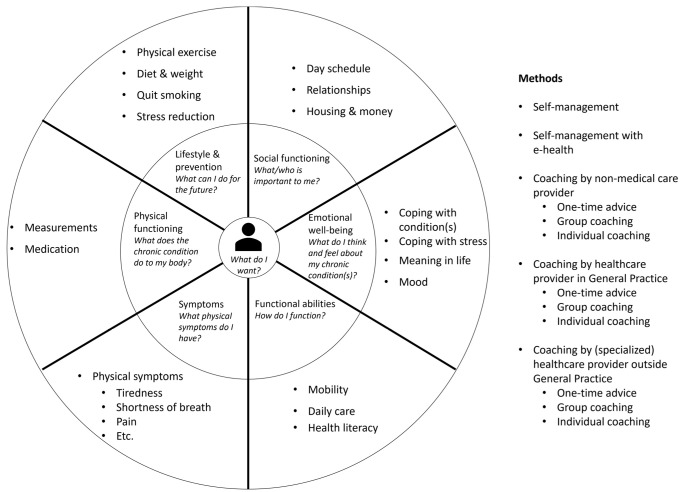
Final intervention model of the Person-Centred and Integrated Care (PC-IC) approach for use in daily practice. This version was adapted from a draft version after the comments of stakeholders in Phases 2 to 4 had been processed.

**Figure 4 ijerph-20-03824-f004:**
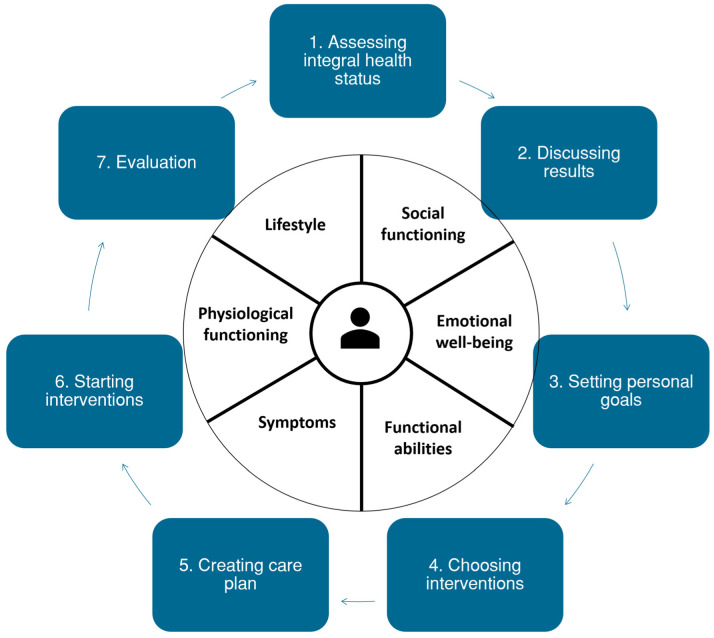
Schematic representation of the Person-Centred and Integrated Care (PC-IC) approach for the management of patients with chronic diseases and multimorbidity in primary care.

**Table 1 ijerph-20-03824-t001:** Overview of the timing and results of the subsequent phases in the development of the patient-centred integrated care approach.

Phase	Population	Time Period	Result
1. Literature review & document analysis	N/A	March–September 2019	Conceptual PC-IC approach
2. Online qualitative surveys	Primary care professionals * purposively selected by the three regional primary care cooperatives involved, supplemented with GPs with a special interest in CVD, DM2, or COPD involved in a national guideline or health policy committees	October–May 2020	Adaptations to conceptual PC-IC approach
3. Interviews	Patients with DM and/or COPD and/or CVD who received chronic disease management from their GP, recruited through the primary care cooperatives involved, by asking involved practices to recruit 1–2 patients from their chronic care population	May–July 2020	Adaptations to conceptual PC-IC approach
4. Finalization	Stakeholders involved in chronic disease care: - Policy advisors of the primary care cooperatives - Policy advisors of healthcare insurance companies - Health literacy experts	June–August 2020	Final version of the PC-IC approach

* from the following professions or disciplines: regular GPs; GPs with a special interest in CVD, DM, or COPD; practice nurses; allied healthcare professionals (e.g., physiotherapists, dieticians); social workers and other healthcare professionals involved in the care of patients with chronic diseases. COPD: chronic obstructive pulmonary disease; CVD: cardiovascular disease; DM2: diabetes mellitus type 2; GP: general practitioner.

**Table 2 ijerph-20-03824-t002:** Description of the thematic data analysis used in the scoping review.

Analysis of the data obtained from the document analysis, online surveys, one-on-one interviews, and focus group interviews was performed using inductive thematical coding, a commonly used method to identify themes in qualitative data. In all phases, at least two authors independently coded the data using ATLAS.ti version 8.4.15. The authors discussed the selected quotations and respective codes after coding one to five documents or transcripts until they reached a consensus. The quotes in the code book were periodically reviewed to check if they needed adaptation, for example, if two codes could be merged together, or if another code should be added. After coding all the documents or transcripts once, the documents were reviewed in light of the latest version of the codebook. After this second review, we categorized the codes using an affinity diagram method. These categories were discussed within the project team. Preliminary results of the analyses were offered for member checking, where possible.

**Table 4 ijerph-20-03824-t004:** Disciplines of the participants in the online qualitative survey study.

	Number
Medical professionals	
GP specialized in COPD, CVD, and/or DM2	8
Regular GP	8
Specialist for elderly care	1
Specialist for internal diseases	1
Nursing professionals	
PN COPD, CVD, and/or DM2	13
PN mental healthcare	2
Home care nurse	2
Allied healthcare professionals	
Dietician	4
Podotherapist	2
Medical pedicure	1
Physiotherapist	1
Pharmacist	1
Other	
Social worker	4
Policy officer of primary care cooperative	3
Lifestyle coach	1
Total	52

Abbreviations: COPD: chronic obstructive pulmonary disease; CVD: cardiovascular disease; DM2: diabetes mellitus type 2; GP: general practitioner; PN: practice nurse.

**Table 5 ijerph-20-03824-t005:** Quotes from the online surveys with healthcare providers (from Phase 2) and individual interviews with patients with DM2, COPD, and/or CVD (from Phase 3).

Quote	Quote	Participant
Healthcare providers		
Q1	‘The holistic view and thinking does not put curing as the highest priority but rather well-being and functioning as desired. A pleasant way for both the caregiver and the patient.’	212
Q2	‘The integral approach can be more time consuming, but will eventually lead to more time, less frustration and more satisfaction.’	201
Q3	‘… giving people back the feeling of control.’	317
Q4	‘It takes a lot of time. Not every healthcare professional is able to do this; not every patient wants to do this.’	324
Q5	‘It will be difficult for patients with limited health skills, while the conversation about integral health status is so important, especially for them.’	206
Q6	‘[Assessment of integral health status] will possibly reveal particular subjects or triggers, which have a negative impact on a patient’s health and would otherwise not have been revealed. For example, financial problems or loneliness.’	319
Q7	‘Some patients might think … some domains are too personal and this might cause resistance in the patient.’	410
Q8	‘Possibly the goals can improve therapy compliance, because the motivation is better.’	207
Q9	‘[Personal goals] can be far away from treatment goals. With high blood pressure, high HbA1c and many cigarettes, it is perfectly possible to take care of grandchildren.’	204
Q10	‘I think it is too detailed, and therefore maybe not usable in practice.’	311
Q11	‘… In addition, it is clear for other involved healthcare professionals what the goals of the patient are and how they want to achieve them. Care can be co-ordinated better.’	306
Q12	‘It could lead to medicalisation of problems which do not originate in the somatic corner. It is quite a lot of work to write it.’	315
Q13	‘Evaluate what worked, also especially what made it work, what didn’t work and what could help to make it work … with a non-judgmental attitude.’	416
Patients with DM2, COPD and/or CVD		
Q14	‘Let’s be honest, you can’t do anything in ten minutes. […] This plan here, […] the practice nurse having half an hour with you, that’s a luxury.’	7
Q15	“You could say to yourself that you are doing alright, but I think that when you have to fill in a questionnaire like that it would make you wonder. How am I really doing?”	8
Q16	“I think that people like myself […] benefit from filling out the questionnaire again after a while and seeing what changes there are, both positive and negative.”	8
Q17	“Some people don’t or don’t want to understand what they are told, because [optimal treatment] means changing their lifestyle, and taking medication may be easier.”	12
Q18	“The more involved your partner is, the better, […] not just for your own support, but also for theirs. They struggle too, sometimes more than you. That is an aspect that is often overlooked.”	7
Q19	“[…]practice nurse, you have to be understanding towards patients that don’t require those strict guidelines.”	12

COPD: chronic obstructive pulmonary disease; CVD: cardiovascular disease; DM2: diabetes mellitus type 2; Q: quote.

## Data Availability

De-identified transcripts of the interviews and survey responses as well as the questionnaires used (all in Dutch) can be made available upon reasonable request.

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
