# Peer review of "Development of a Person-Centred Integrated Care Approach for Chronic Disease Management in Dutch Primary Care: A Mixed-Method Study"

_ijerph, 2023, doi:10.3390/ijerph20053824_

Round 1

Reviewer 1 Report

The authors present the results of a multiphase study to develop a new person-centred integrated care (PC-IC) model to improve care for patients with multiple chronic diseases in primary care. They justify the need for this new care model based on the fact that the Disease Management Programme (DMPs) currently available and focusing on a single disease approach is not align with patients’ needs and primary care practices competencies.

This mixed-method study has been executed from March 2019 to July 2020 in 3 regions of the Netherlands, and follow 4 phases: 1/ scoping / documents review to construct the model; 2/ online survey with experts to refine the model; 3/ qualitative survey with patients to refine the model and 4/ final refinement with other stakeholders. Implication of all stakeholders, including patients, HCP and administrators/health insurance, is to be

The study is relevant and interesting, results proved insight on how primary care practices could evaluate to better respond to the needs of patients with multiple chronic diseases. The article is well written and the 4 phases are relevant to answer the question studied. The 4 phases complement each other and flow well throughout the article. Nevertheless, there is a lack of details on the method, perhaps due to the presence of the different phases to be described in a reduced format. Authors do not comply with existing reporting guidelines to report on qualitative research or online surveys, I understand that this is not feasible for all stages of the study, but it would be necessary to specify certain points to evaluate the quality of the project's conduct.

Introduction:

- How are DMP implemented and funded in daily practices in Dutch PC?

- Introduction would benefit from a conceptual definition of the person-centered approach and of the integrated care approach  

Methods

- General comment: population inclusion criteria and recruitment are poorly described for phases 2-4.

-2.2 Scoping review method: what definition of multiple chronic condition was used? Were there publication dates for limits in the scoping review search?

- 2.3 Online qualitative survey: the method used by the author is not a traditional/ qualitative inquiry method but rather an online survey with open ended responses, with a thematic analysis of wordings. The method used does not allow for a thorough exploration of the HCP perspectives. A 5 round process is described, but it is not clear how the successive rounds were articulated, were there decision rules at each stage of the process? Is it possible to provide the translated questionnaires as supplementary material?

- 2.4: What were the inclusion criteria, the sampling approach? Did patients had to have at least two chronic diseases? The interview guide with patients show that patients were asked to compare the model to usual care. What is their ‘usual care’? Is it the DMP that is described in the introduction of the paper? (appendix B)

-2.5 This part is not very well structured; it does not follow standardized objectives and methodology. Did the organization of the meetings follow a specific approach to collect feedback from the participants? Who presented the results and facilitated the exchanges with the participants? Were patients invited at this stage?

Results:

Figure 2: Which results are supporting the use of digital tools to conduct the assessment?

It is not clear to me to what the “tool and overview of treatment possibilities” refers and how it is developed ?

3.1.2 Figure 3 does not identify what comes from the analysis of the documents and what comes from the comments of stakeholders. I would suggest to refer to figure 3 at the end of the process since it is an output of the 4 phases. In the 3.1.2 section, main components that have been defined based on the document analysis may be briefly described by text?  

The broad outlines of the content and stages of the model are described but it seems throughout the manuscript that tools or approaches are to be deployed at each stage (questionnaire, care plan intervention choice tool, shared decision). Have these elements been created, detailed and presented to the participants of the different stages? In the results, it is sometimes difficult to understand what is being reported:  

- Does the model include a method to elaborate the care plan? To conduct SDM?

- 3.2.2 “Using digital questionnaires was appreciated by the participants”  to what questionnaire does this refer?

- Lines 571 – 572 : A digital questionnaire was tested: which questionnaire ? 

- 3.4.2 This section is difficult to understand: what has been done/tested with who?

- Throughout the results of the patients interviews, patients assumptions are not supported by quotes. May the authors add an additional table with patients quotes as done for HCP?  

Discussion

- Regional differences are discussed but not presented in the results section?  Did the authors retrieved different results in the 3 regions? If so it has to be presented in the results before the discussion, or it has to be deleted from the discussion.

- Several models of primary care practices models have been proposed internationally, for instance; the Patient Centered Medical Home in the US and Patient’s Medical Home in Canada, it may be interesting to discuss how the developed model for PC-IC fit with the pillars of those models and of the core competencies of family physicians?

- The limitation section has to be expanded, notably with methodological concerns about the number of participants and their characteristics for the interviewed patients: the inclusion of 8 males on 9 interviewees strongly limits the study's ability to take gender into account in the model. This reduced diversity of the population may as well explain why saturation has been reached so quickly, as 8 participants to reach saturation is unusually early. Generalisability of results to other NL regions / countries might also be discussed in this section.

- Implication for research and practice may be deepen. Brakes and enablers to the implementation of the model, as well as organizational changes, must be discussed and anticipated for the next study that is presented, and links with health literacy and shared decision making skills development could be done.

Minor points

Line 681 : ref to be included

Appendix C : title ?

Reviewer 2 Report

In the present manuscript entitled “Development of a person-centred integrated care approach for chronic disease management in Dutch primary care: a mixed-method study” the authors have performed a person-centred and integrated care approach for people with chronic conditions in primary care. Overall, the present manuscript is well-written and sound.

Reviewer 3 Report

This paper describes the development of a person-centered, integrated care approach for the management of chronic disease in the Dutch primary care arena.

This study is very important because it presents a new concept of how to approach the multimorbidity patient.

Methodologically, the author uses multiple qualitative research methods. So this study tends to be arbitrary, however, I do not think the value of this study is lost.

The paper would be even better if the following five points were improved

1.       In the abstract, I don't think the author clearly states the conclusion. The author should clearly state what the PC-IC approach is in the part before "Future evaluation".

2.       In Table 1, the authors describe four phases. If the author has a reason for planning to conduct the study in these four phases, please provide the reason for choosing the four phases in the text.

3.       In writing the paper, I would like to give some general notes. In Table 4, for abbreviations such as GP, COPD, etc., please state what the abbreviation means so that it can be understood from the table alone.

4.       In the "Discussion" section, in the "4.1 Summary of Results" part, the following statement is made "Added a patient-level evaluation-step to the PC-IC approach." Does this mean that the third of the four phases was added after the study was initiated? Or do you mean that you added a patient-level evaluation-step within the PC-IC approach? Please clarify which one you mean.

5.       In the "5. Conclusions" section, I don't think the author clearly states the conclusion. The author should clearly state what the PC-IC approach is in the part before "Future evaluation". This proposal has the same intention as Proposal 1.

Round 2

Reviewer 3 Report

The author has revised the five points I suggested. I believe the manuscript is an improvement.